# The Outcomes of Liver Transplantation in Severe Metabolic Dysfunction-Associated Steatotic Liver Disease Patients

**DOI:** 10.3390/biomedicines11113096

**Published:** 2023-11-20

**Authors:** Natasa Paklar, Maja Mijic, Tajana Filipec-Kanizaj

**Affiliations:** 1Department of Anesthesiology, Reanimatology and Intensive Care, University Hospital Merkur, 10000 Zagreb, Croatia; 2Department of Gastroenterology, University Hospital Merkur, 10000 Zagreb, Croatia; 3School of Medicine, University of Zagreb, 10000 Zagreb, Croatia

**Keywords:** non-alcoholic steatohepatitis, non-alcoholic fatty liver disease, liver transplantation, metabolic syndrome, outcomes

## Abstract

The increasing prevalence of diabetes mellitus, obesity, and metabolic syndrome in the population can lead to metabolic dysfunction-associated steatohepatitis (MASH) and metabolic dysfunction-associated steatotic liver disease (MASLD). In Western industrialized countries, this has become a major problem with significant socioeconomic impacts. MASH is now a leading cause of liver transplantation (LT), especially in developed countries. However, the post-transplant outcomes of such patients are a major concern, and published data are limited and extremely variable. In this article, we discuss graft and patient survival after LT, complications, the recurrence of MASH, and MASH appearing de novo after transplantation. Recent studies suggest that patients with MASH have slightly worse short-term survival, potentially due to increased cardiovascular mortality. However, most studies found that longer-term outcomes for patients undergoing LT for MASH are similar or even better than those for other indications. Hepatocellular carcinoma due to MASH cirrhosis also has similar or even better outcomes after LT than other etiologies. In conclusion, we suggest questions and topics that require further research to enhance healthcare for this growing patient population.

## 1. Introduction

More than 70 years ago, the association of hepatic steatosis with inflammatory changes and fibrosis in obese patients was described for the first time [1]. Non-alcoholic steatohepatitis (NASH) as a special etiology category was added to transplant databases in the United States of America (US) in 2001. In the last two decades, non-alcoholic fatty liver disease (NAFLD) has evolved into the second highest cause of liver transplantation (LT) in the US [2]. The term NAFLD was first used in a review article by Schaffner and Thaler in 1986 and the term NASH was first used by Ludwig in 1980 [3,4].

In June 2023, a new nomenclature was adopted by experts from more than 50 countries. This was implemented under the auspices of the American Association for Study of Liver Disease (AASLD) and the European Association for Study of the Liver (EASL) and in cooperation with the Asociación Latinoamericana para el Estudio del Hígado (ALEH), and is now the official terminology [5]. The entity of metabolic dysfunction-associated steatotic liver disease (MASLD) encompasses patients who have hepatic steatosis and have at least one of five cardiometabolic risk factors. A new category, outside pure MASLD, termed MetALD was selected to describe those with MASLD who consume greater amounts of alcohol per week (140 g/week and 210 g/week for females and males respectively). From a histological standpoint, those with no metabolic parameters and no known cause have cryptogenic steatotic liver disease. From a histological standpoint, MASLD can range from simple steatosis of the liver (fatty liver and steatohepatitis (MASH, formerly NASH)) to liver fibrosis, which can lead to the terminal stage of liver disease (cirrhosis) with high morbidity and mortality rates [6,7,8]. 

MASLD in most instances represents a so-called benign condition with fat infiltration in the liver but without inflammation. MASH is fat infiltration in the liver with inflammatory changes [9]. To diagnose MASH, the minimum histological criteria include the presence of cytological ballooning, lobular inflammation, and hepatic steatosis. Further progression can lead from MASH to further fibrosis and cirrhosis, and consequently, to further liver failure and hepatocellular carcinoma (HCC) [7,8]. Subsequently, cirrhosis may lead to HCC [10]. Unfortunately, MASLD patients, even without liver cirrhosis, can develop HCC. Today, it is the most common ethology of HCC in many countries [11]. 

Recently, instead of the name MASLD, the name MAFLD was coined to describe fatty liver disease associated with metabolic syndrome [12,13]. MASLD is considered a hepatic manifestation of metabolic syndrome (MS), which represents a series of abnormalities associated with insulin resistance. Central obesity is one of the major components [14]. For MAFLD, it is enough to have liver steatosis determined histologically, via imaging methods or serological markers in patients who are overweight or/and have diabetes mellitus (DM) type 2 or other signs of metabolic dysfunction: increased abdominal girth, hypertension, insulin resistance, or impaired glucose tolerance [13]. It is believed that environmental factors, genetics, and insulin resistance are the primary risk factors for developing MASLD [15,16]. Insulin resistance is a prevalent observation among patients who have MASLD [17,18,19,20]. In industrialized Western countries, the prevalence of obesity, DM, and MS has made MASLD a significant concern [21,22,23,24,25,26,27,28]. The complex process that leads from MASLD to cirrhosis is not completely defined [29,30].

Henceforth in this paper, NAFLD will be referred to as MASLD and NASH as MASH.

As was mentioned earlier, MASH has become a growing reason for liver transplant (LT) in developed countries, which raises questions about post-transplant outcomes. 

The purpose of this paper is to provide a brief insight into the history of MASH, epidemiology, trends in LT, complications, survival, and problems, as well as future perspectives.

## 2. Epidemiology

MASLD is a prevalent condition affecting a significant portion of the population in several countries, and the actual number of those affected is greater than previously assumed. According to recent research, the prevalence of MASLD has increased from 25.5% before 2005 to 37.8% after 2016 [31]. With the rise of obesity and the adoption of sedentary, unhealthy lifestyles in developing countries, we are witnessing a global epidemic of obesity, diabetes, high blood pressure, and hepatitis. As a result, we can expect an increase in the prevalence of MASLD [32]. It has been reported that there is a significant correlation between the occurrence of MASLD and obesity [33]. The United Network of Organ Sharing (UNOS) reported the first liver transplant in the US for MASH in 1996 [7]. From 1995 to 2000, the percentage of liver transplants due to cirrhosis caused by MASH was 0.1%. By 2005, it had increased to 3.5% [7]. On the other hand, new highly effective antiviral regimens and early virus detection have resulted in fewer cases of cirrhosis caused by viral infections. MASH stands as the second leading cause of liver transplant on waiting lists in the United States of America (US), and it is the primary cause in women in the US [34,35]. MASH emerged in 2016 as the primary indication for liver transplantation in the US among individuals born between 1945 and 1965 [36]. A recent study found that MASH has surpassed other causes as the primary etiology in transplants for the elderly population over 65 between 2018 and 2020 [37]. The dramatic increase in MASLD has significant economic and public health implications that cannot be ignored [32,38].

## 3. Outcomes

Reviewing the literature, one can come across a couple of studies addressing the outcomes of LT for MASH cirrhosis (Table 1).

The most valuable analysis is of large official databases, but no less valuable is a single-center study that emphasizes special problems related to MASH cirrhosis (e.g., distinguishing MASH cirrhosis from cryptogenic cirrhosis (CC)...). All but one multicentric studies were conducted in the US. Only one multicentric study is from Europe. 

**Table 1 biomedicines-11-03096-t001:** Outcomes of liver transplantation in severe MASLD patients.

Study	N	Survival	MASH	Non-MASH	CC	HCV	HBV	AIH	ALD	PSC
Large database studies										
Charlton et al. [36] Period: 2001–2009 Data source: SRTR	35,781 LT; 1959 MASH	1 y	84%	87%	86%	•	•	•	•	•
		3 y	78%	78%	79%	•	•	•	•	•
Karnam et al. [39]. Period: 2002–2019 Data source: SRTR	6515 LT for MASH cirrhosis	5 y	79%	•	•	•	•	•	•	•
Younossi et al. [40]. Period: 2002–2016 Data source: SRTR	158,347 LT candidates due to HCC	1 y (mortality)	10.6%	•	•	10.6%	8.1%	8.6%	•	•
		3 y (mortality)	19.7%	•	•	24.4%	16.7%	19.9%	•	•
		5 y (mortality)	28.2%	•	•	34.9%	21.5%	31.3%	•	•
Afzali et al. [41] Period: 1997 to 2010 Data source: UNOS	69,962 LT; 1810 were MASH recipients	1 y	87.6%	•	•	•	•	•	•	•
		3 y	82.2%	•	•	•	•	•	•	•
		5 y	76.7%	•	•	•	•	•	•	•
Singal et al. [42] Period: 1994–2009 Data source: UNOS	54,687 LT; 1358 were due to MASH cirrhosis	1 y	88.8%	86.9%	87.2%	89.5%	•	•	88.6%	93.4%
		3 y	85.4%	82.4%	79.9%	84.5%	•	•	83%	89.7%
		5 y	84.1%	78.6%	75.9%	82.4%	•	•	79.4%	87.4%
		10 y	84.1%	78.6%	75.9%	82.4%	•	•	79.4%	87.4%
Thuluuath et al. [43]. Period: 2002–2019 Data source: UNOS	Cryptogenic (3241 patients) and MASH (4089 patients) cirrhosis	30 d	97%	•	96%	•	•	96%	97%	•
		1 y	89%	•	87%	•	•	88%	90%	•
		2 y	86%	•	85%	•	•	86%	88%	•
		3 y	83%	•	82%	•	•	84%	84%	•
		5 y	77%	•	77%	•	•	79%	78%	•
		10 y	63%	•	61%	•	•	65%	60%	•
Cholankeril et al. [44]. Period: 2003–2014 Data source UNOS/OPTN	63,061 LT, including 20782 HCV (32.96%)	5 y	77,81%	•	•	72,15%	•	•	•	•
Rajendran et al. [45]. Period: 2001–2009 Data source: UNOS	35,781 LT, of which 1959 were MASH	1 y	84%	87%	•	•	•	•	•	•
		3 y	78%	78%	•	•	•	•	•	•
Kwong et al. [46]. Period: 2010–2016 Data source: REALT	1023 LT, of which 207 (20.2%) were due to MASH cirrhosis	1 y	91.3%	90.1%	•	•	•	•	•	•
		3 y	83.3%	81.5%	•	•	•	•	•	•
Haldar et al. [47] Period: 2002–2016 Data source: ELTR	68,950 LT and 2741 MASH recipients	1 y	84.1%	86.2%	•	•	•	•	•	•
		2.5 y	80.2%	81.6%	•	•	•	•	•	•
		5 y	73.4%	75.4%	•	•	•	•	•	•
		10 y	62.1%	62.9%	•	•	•	•	•	•
Jamil et al. [48]. Period: 2005–2019 Data source: SRTR	Over 80,000 LT	10 y	61%	•	•	•	•	•	•	•
Nagai et al. [49]. Period: 2016–2018 Data source: OPTN/UNOS	6344 LT for MASH, 17,037 for HCV, and 9279 for ALD	1 y	90.4%	•	•	92.8%	•	•	•	•
Single-center studies										
VanWagner et al. [50]. Period: 1993–2010 Data source: Northwestern Memorial Hospital and the University of Chicago Medical Center	115 had MASH (or CC with known risk factors for MASH); 127 patients with alcohol-induced cirrhosis	1 y	81.3%	•	•	•	•	•	88.1%	•
		3 y	73.3%	•	•	•	•	•	85.3%	•
		5 y	60.3%	•	•	•	•	•	68.8%	•
Kennedy et al. [51]. Period: 1999–2009 Data source: University of Alabama at Birmingham	129 recipients with MASH and 775 recipients were non-MASH	1 y	90%	92%	•	•	•	•	•	•
		3 y	88%	86%	•	•	•	•	•	•
		5 y	85%	80%	•	•	•	•	•	•
Malik et al. [52]. Period: 1997–2008 Data source: University of Pittsburgh School of Medicine	2021 LT; 98 patients with MASH cirrhosis	24 h mortality	4.1%	1–3%	•	•	•	•	•	•
		30 d mortality	6.1%	2–5%	•	•	•	•	•	•
		2 y mortality	21.4%	13–18%	•	•	•	•	•	•
		3 y mortality	25.5%	16–30%	•	•	•	•	•	•
		5 y mortality	27.6%	19–35%	•	•	•	•	•	•
Bhagat et al. [53]. Period: 1997–2007 Data source: University of Miami	For CC with the MASH phenotype (71 patients) or alcoholic cirrhosis (83 patients)	1 y	82%	•	•	•	•	•	92%	•
		3 y	79%	•	•	•	•	•	86%	•
		5 y	75%	•	•	•	•	•	86%	•
		9 y	62%	•	•	•	•	•	76%	•
Sadler et al. [54]. Period: 2004–2014 Data source: University of Toronto and University of California San Francisco	929 LT were due to HCC and 60 were due to HCC in MASH cirrhosis	1 y	98%	95%	•	•	•	•	•	•
		3 y	96%	84%	•	•	•	•	•	•
		5 y	80%	78%	•	•	•	•	•	•
Agopian et al. [55]. Period: 1993–2011 Data source: The University of California, Transplant and Liver Cancer Centers in Los Angeles	144 adult MASH patients	1 y	84%	•	•	•	•	•	•	•
		3 y	75%	•	•	•	•	•	•	•
		5 y	70%	•	•	•	•	•	•	•
Barrit et al. [56]. Period: 2004–2007 Data source: University of North Carolina Hospital	118 LT, and 18% were due MASH cirrhosis	30 d	81%	95%	•	•	•	•	•	•
		1 y	76%	83%	•	•	•	•	•	•
		3 y	76%	73%	•	•	•	•	•	•
El Atrache et al. [57]. Period: 1996–2008 Data source: Henry Ford Hospital, Detroit	MASH (46 patients) and CC (37 patients)	10 y	80%	•	•	•	•	•	•	•
Bhati et al. [58]. Period: 1995–2013 Data source: University of Maryland School of Medicine, Baltimore	103 LT; 48 had MASH cirrhosis	5 y	86%	•	•	•	•	•	•	•
		10 y	71%	•	•	•	•	•	•	•
		15 y	51%	•	•	•	•	•	•	•
Sanjeevi et al. [59]. Period: 2016–2018 Data source: University of Nebraska Medical	71 MASH patients, and 6 of them had HCC	1 y	87.6%	•	•	•	•	•	•	•
		3 y	82.2%	•	•	•	•	•	•	•
		5 y	76.7%	•	•	•	•	•	•	•
Holzner et al. [60]. Period: 2001–2017 Data source: School of Medicine at Mount Sinai, New York	635 LT, of which 51 (8%) were MASH-HCC	1 y	92%	•	•	86%	93%	•	88%	•
		3 y	86%	•	•	76%	87%	•	76%	•
		5 y	80%	•	•	65%	83%	•	69%	•
Kakar et al. [61]. Period: 2000–2015 Data source: University of Pittsburgh	226 patients with MASH	1 y	82%	•	•	•	•	•	•	•
		5 y	73%	•	•	•	•	•	•	•
		7 y	62%	•	•	•	•	•	•	•
Yalamanchili et al. [62]. Period: 1986–2004 Data source: Baylor University Medical Center, Dallas	2052 LT; 7% for MASH (the cohort of MASH patients were with CC)	1 y	85.6%	86.3%	•	•	•	•	•	•
		5 y	71.4%	69.9%	•	•	•	•	•	•
		10 y	56.5%	52.7%	•	•	•	•	•	•
		20 y	12.6%	20.6%	•	•	•	•	•	•
Kern et al. [63]. Period: 2002–2012 Data source: Medical University of Innsbruck, Austria	513 LT; 12.7% for MASH cirrhosis	1 y	93.2%	•	•	•	•	•	•	•
		3 y	78.5%	•	•	•	•	•	•	•
		5 y	72.1%	•	•	•	•	•	•	•
Castello et al. [64]. Period: 1997–2016 Data source: La Fe University Hospital, Valencia	1986 LT; 40 (2%) were labelled as MASH-related	1 y	89%	•	•	•	•	•	83%	•
		3 y	89%	•	•	•	•	•	78%	•
		5 y	83%	•	•	•	•	•	72%	•
Heuer et al. [65]. Period: 2007–2011 Data source: University Hospital of Essen, Essen, Germany	432 LT; 40 due MASH-induced cirrhosis	4 y mortality	60%	•	•	•	•	•	•	•
Tokodai et al. [66]. Period: 2007–2017 Data source: Karolinska University Hospital, Stockholm, Sweden	694 LT; 27 MASH patients; and 68 ALD patients	1 y	89%	•	•	•	•	•	91%	•
Tanaka et al. [67]. Period: 1996–2013 Data source: Japan	425 living-donor LT; 7 due to MASH	5.3 y	100%	•	•	•	•	•	•	•
Jothimani et al. [68]. Period: 2009–2019 Data source: Bharath Institute of Higher Education and Research; India	1017 LT, of whom 396 had MASH cirrhosis	1 y	86.6%	•	•	91.3%	93.5%	•	86%	•
		3 y	81.8%	•	•	86.1%	88.5%	•	82.9%	•
		5 y	75.9%	•	•	86.1%	88.5%	•	79.7%	•

MASH—metabolic dysfunction-associated steatohepatitis; MASLD—metabolic dysfunction-associated steatotic liver disease; UNOS—United Network of Organ Sharing; HCC—hepatocellular carcinoma; LT—liver transplantation; ILTS—International Liver Transplantation Society; CC—cryptogenic cirrhosis; SRTR—Scientific Registry of Transplant Recipients; HBV—hepatitis B virus; AIH—autoimmune hepatitis; PSC—primary sclerosing cholangitis; ALD—alcohol liver disease; PBC—primary biliary cirrhosis; OPTN—Organ Procurement and Transplantation; REALT—Re-Evaluating Age Limits in Transplantation; ELTR—European Liver Transplant Registry; •—not recorded.

### 3.1. Large Database Studies

In most studies, MASH was comparable with other indications of LT including CC; hepatitis B virus, HBV; autoimmune hepatitis, AIH; primary sclerosing cholangitis, PSC; and alcohol liver disease, ALD [28,35,37,38,39]. On the other hand, some studies suggest that survival rates for MASH cirrhosis are better than for certain indications for LT, but worse than others [41,42,44].

Two studies have also been published that indicate poorer survival for recipients and transplants with MASLD. One of the recent studies by Jamil et al. points out that recipients with MASH cirrhosis are older, so the expected length of life for such patients is poorer [48]. Nagai et al. also found that the risk of death within 1 year after transplant was higher among patients with MASH than those with HCV-associated liver disease or ALD [49]. The risk of death increases with age, and patients with MASH have a higher risk of death from cardiovascular or cerebrovascular disease; MASH patients also have an increased risk of post-transplant mortality compared with those with hepatitis C.

Several studies have demonstrated that liver transplant recipients with MASH tend to have specific characteristics that distinguish them from those with other indications for liver transplants. MASH recipients are generally older, have a higher BMI, and are more likely to be female and of White or Hispanic ethnicity. They also tend to have a higher model for end-stage liver disease-sodium score, and a lower frequency of HCC [36,45,46].

### 3.2. Single-Center Studies

Most of the single-center studies were also conducted in the US. Few were conducted in Europe, one was conducted on live donors in Japan, one was conducted in India, and none were found from other regions.

Liver transplant survival rates vary greatly among different studies. While most studies show comparable or positive results for MASH, studies conducted at single centers tend to report poor outcomes [56,65,66].

A group of authors who followed Malik reported that early mortality in MASH recipients was increased, but the 5-year mortality was similar to patients who underwent transplantation for other indications [52].

The situation is similar with HCC in MASH cirrhosis [10,54,60].

Due to the specific situation in Japan, Tanaka et al. conducted a study that solely focused on living-donor liver transplantation [67].

## 4. Common Complications

### 4.1. Cause of Death

The leading causes of long-term mortality were infection (15.9–38%), cardiovascular events (5.3–26%), multiorgan failure, graft failure (6–41%), and malignancy (liver as well as non-liver malignancy; 2.5–9.3%) [41,43,47,51,56,58,69]. However, patients with MASH were at lower risk of graft failure compared with patients without MASH [70,71].

### 4.2. Recurrent MASLD

Post-transplantation MASLD/MASH can be categorized into two subgroups: recurrent and de novo. MASLD is a common complication in recipients, irrespective of transplantation indication, with a prevalence ranging between 8% and 100% in a follow-up period of 1–10 years, but it mainly did not progress to fibrosis and it is rarely an indication for retransplantation [60,61,62,69,72,73,74,75,76,77]. Post-LT MASLD/MASH is often underdiagnosed due to the poor sensitivity of most routine imaging methods.

### 4.3. Metabolic Syndrome

Immunosuppressive agents can exacerbate a pre-existing MS in recipients or lead to a de novo MS [71]. Corticosteroids increase the hepatic output of glucose and decrease insulin production and peripheral glucose uptake. Corticosteroid use has been associated with an increased risk of DM type 2, dyslipidemia, hypertension, and rapid weight gain in recipients following LT [78]. Calcineurin inhibitors also represent a risk for MS and consequent post-LT MASLD. They are linked to hypertension (mainly associated with cyclosporine), dyslipidemia, new-onset DM type 2 (with tacrolimus having a more prominent diabetogenic effect), and chronic renal disease [79,80,81,82].

In Yalamanchili et al.’s survey, which mainly reviewed complications after LT, 14% of MASH cirrhosis recipients had lost 10 kg of their baseline weight, suggesting that their initial dry body weights may have been overestimated because of the presence of ascites and edema [62]. New-onset diabetes developed in 35.8% of those at risk (no pretransplant diabetes). New-onset hypertension requiring medication developed in 61.5% of those at risk. The new onset of diabetes or hypertension was unrelated to the initial immunosuppressive regimen.

### 4.4. Thrombosis and Cardiovascular Events

Underlying MASH is associated with cardiovascular events. Cardiovascular events occurred in 26% of the patients with confirmed MASH post-transplant at 5 years [69]. There were no differences in cardiac event occurrence when comparing MASH and ALD at 1 year (7.7 vs. 6.1%) and 3 years after LT (14.1 vs. 13.8%) [83]. In 5.1% to 6.7%, death was attributed to pulmonary embolisms [43]. VanWagner et al. noticed that MASH patients were more likely to experience an adverse cardiovascular event in the first year after LT compared to alcohol cirrhosis patients [50]. The most common cardiac complication in both groups was acute pulmonary edema (18.1% MASH versus 16.2% alcohol cirrhosis), followed by new-onset atrial fibrillation (10.3% MASH versus 8.4% alcohol cirrhosis).

### 4.5. Malignancy

The rates of malignancy within 1 year were also similar between MASH and non-MASH recipients, with an incidence of 4.9% for solid-organ malignancy (of which 40% was recurrent HCC), 3.8% for skin cancer, and 1.3% for post-transplant lymphoproliferative disease [46]. There was no difference in the number of tumor recurrences nor the frequency of recurrence location between groups (13.3–87.5% in the MASH group vs. 14–50.8% in the non-MASH group) [2,54,84].

### 4.6. Infection

Over 50% of all the deaths in the MASH cohort were the result of infection [52]. The majority of septic complications were observed in the early postoperative period [69].

Gitto et al. explored the correlation between MS and renal dysfunction post-LT in patients with MASH [85,86]. Post-LT diabetes, dyslipidemia, and hypertension can worsen renal function in these patients.

Kwong et al. found that there was no significant difference in the incidence of these complications between MASH and non-MASH recipients [46]. Cardiovascular, neurological, and infectious outcomes were similar between the two groups, including atrial fibrillation (13.7%), myocardial infarction (MI; 2.8%), stroke (5.5%), heart failure (6.3%), delirium (12.5%), seizures (4.2%), viral infection (15.9%), bacterial infection (36.6%), and fungal infection (5.7%). Renal outcomes were also comparable, with 4% of LT recipients on dialysis at 3 months and 1.5% on dialysis at 12 months. MASH LT recipients had a lower median eGFR at each recorded time point.

### 4.7. Therapy

There is no approved pharmacotherapy for MASLD specific to transplanted patients. The application of therapy is based on guidelines for the general population of patients with MASLD. Lifestyle modification and weight loss remain the cornerstones of MASLD treatment. There are currently no FDA-approved medications for the specific treatment of MASLD. Drugs approved to treat associated comorbidities with potential benefits in MASLD may be considered in the appropriate clinical setting. These include pharmacotherapy with pioglitazone or semaglutide in diabetics, vitamin E in non-diabetic patients with MASH, and bariatric surgery. In the context of transplantation, it is important to reconsider the treatment of metabolic alterations related to immunosuppression. For the optimization of associated metabolic comorbid disease, a multidisciplinary team of clinicians provides the best chance for success in reducing liver and cardiovascular morbidity and mortality in patients with MASLD.

## 5. Future Perspectives and Conclusions

Today, MASLD is becoming the leading indication for LT, surpassing viral hepatitis in recent years. This is a result of an increase in the overall prevalence of obesity and metabolic syndrome, coupled with the reduction in chronic hepatitis C infection and better antiviral treatment. There are concerns about the post-transplant survival of patients with MASLD because of associated cardiovascular and metabolic risk factors [85].

There are no clear findings on this issue because the results of the performed studies are ambiguous. The above is significantly related to the characteristics of studies in which the outcomes of liver transplantation in MAFLD patients were examined in relation to other indications.

Emerging evidence suggests that the short-term survival of transplanted MAFLD patients may indeed be slightly worse and that this may be due to increased cardiovascular mortality. Some other studies are suggesting poorer outcomes explained by different reasons (e.g., differences in baseline characteristics and long follow-up periods) [48,50,69]. Most of the other studies found that outcomes for patients undergoing LT for MAFLD are similar or even favorable to other indications. Also, HCC due to MAFLD cirrhosis has similar or even favorable outcomes to those for other etiologies.

The main problem with finding good-quality studies is that most studies available today are retrospective, opening up the question surrounding the overlapping of multiple etiologies of liver disease, especially alcoholic and non-alcoholic. Moreover, it is believed that a significant proportion of patients diagnosed with CC likely represent cases of unrecognized MAFLD, especially those in earlier years.

It is also not clear whether differences in liver transplant outcomes between MAFLD and other indications can be explained by differences in recipients’ baseline characteristics or by other parameters related to the pathogenesis of the underlying disease and its associated comorbidities. Most of the studies showed that patients undergoing LT for MASLD cirrhosis are older, have a larger BMI, and are more likely to be female.

An additional challenge is the design of the studies. The outcome of transplantation and the appearance of complications, as well as the timing of their occurrence, are influenced by various factors that may not be related to the underlying disease etiology. As Sanyal et al. emphasize, shorter periods are not adequate for slowly progressive diseases such as MASLD [87]. Even with a longer follow-up period, patient cohorts must be matched for basic demographic characteristics. As mentioned earlier, MAFLD patients are older and expected to have shorter life expectancy.

It was seen from the studies that even different time frames make a difference. As it turned out, some studies studying the same database but with only a difference in a period of one or two years gave different results [41,42].

In order to overcome this, in further studies, recipients should be clearly stratified by baseline characteristics and with a longer follow-up period.

Equally important are studies investigating the course of MASLD after transplantation and the associated risk factors. The above leads to the question of what the optimal method for monitoring changes in the graft and other organs associated with MASLD is. The biopsy is currently the gold standard for diagnosing severe liver disease. However, non-invasive methods for this population are being researched. Identifying risk factors for the recurrence of the disease in both donors and recipients, along with biomarkers that predict liver-related outcomes, can help in identifying patients who may benefit from treatment. Furthermore, non-invasive diagnostic markers can play a crucial role in timely diagnosis and treatment, allowing for better patient stratification based on risk.

Many drugs have been tested for the treatment of MASH in non-transplant settings (e.g., pioglitazone, metformin, vitamin E, pentoxyphylline, ursodeoxycholic acid, glucagon-like peptide-1 receptor agonist, and sodium–glucose cotransporter inhibitors). Unfortunately, no controlled studies have been conducted in a population of transplanted patients and a directed approach to this specific population is not included in the guidelines. Therefore, the approach is based on recommendations for the general population and the treatment of metabolic complications associated with MASLD. Lifestyle modification remains the cornerstone of management [88]. Weight loss and increased physical activity are effective mediators of MASLD, and their role in cardiovascular risk reduction is well established. The above represents one of the important areas of future research in which it is necessary to investigate a directed approach that respects the specificities of transplanted patients and the effects of immunosuppression.

## Data Availability

Data are contained within the article.

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
