# Peer review of "The Outcomes of Liver Transplantation in Severe Metabolic Dysfunction-Associated Steatotic Liver Disease Patients"

_biomedicines, 2023, doi:10.3390/biomedicines11113096_

Round 1
Reviewer 1 Report
Comments and Suggestions for Authors
Dr. Natasa Paklar et al. have reported that the review of outcome of liver transplantation for non-alcoholic steatohepatitis (NASH). This is interesting and very important report for hepatologist and general practitioners.
I have some questions and comments, however, that I believe the authors need to address, as follows:
<Major criticisms>
1. The authors summarize results of many papers well. However, the final author's message is not conveyed. They need to clarify this point.
2. The prognosis for liver transplantation in young patients with NASH is poor, according to Alkhouri N [44] et al. This paper differs from the others in that it deals primarily with pediatric cases. The present review deals mainly with adult cases, and it is questionable whether it is appropriate to add the article by Alkhouri N to the discussion. The authors need to consider this point in detail.
3. Recently, Jie Ning Yong et al. reported the outcome of liver transplantation for NASH in Clinical Gastroenterology and Hepatology 2023;21:45-54. The authors should show us about the difference with this paper.
<Minor criticisms>
1. The text is littered with unnecessary period after the year, etc. The following period should be omitted.
l On page 1 line 33, ‘1986.’
l On page 3 line 102, ‘cholangitis.’
l On page 3 line 105, ‘2019.’
l On page 3 line 114, ‘2002-2016.’
l On page 3 line 122, ‘1997. to 2010.’
l On page 3 line 126, ‘1994. -2009.’
l On page 3 line 149, ‘1987. to 2012.’
etc.
2. On page 2, line 49, DM abbreviation needs to be explained.
3. In the sentence, a period should be a comma. This needs to be corrected.
l On page 2 line 80, ‘0,1%’
l On page 2 line 81, ‘3,5%’
l On page 3 line 150, ’6,6 years)’
l On page 4 line 151, ‘11,5%’
etc.
4. On page 3 line 121, the hyphen in ‘(UNOS)-’ is not required.
5. On page 3 line 148, the hyphen in ‘com-plicated’ is not required.
6. On page 5 line 235, the ‘Dumont-‘ is not required.
7. Please confirm that this is correct for 'NAFLD' on page 5 line 246 and 247.
8. On page 5 line 254, ')' is required after 'pre-transplant'.
9. On page 6 line 269, ‘NAFL’ should be ‘NAFLD’.
10. On page 9 line 433, ‘86’ should be ‘[86]’.
Author Response
Thank you for giving us the opportunity to submit a revised draft of our manuscript titled ‘Outcomes of liver transplantation in severe MASLD patients’ to Biomedicines (Special Issue: NASH and Hepatocellular Carcinoma (HCC)). We appreciate the time and effort that you dedicated to providing your valuable feedback on our manuscript. We are grateful for Your insightful comments on our paper. We addressed all the concerns you raised and have been able to incorporate changes to reflect most of the suggestions you provided. We wish to highlight that the new nomenclature was utilized in the revised version of article, in accordance with the agreement reached with the editors. Here is a point-by-point response to your comments and concerns:
<Major criticisms>
- The authors summarize the results of many papers well. However, the final author's message is not conveyed. They need to clarify this point.
Response: Thank you for pointing this out. Therefore, we have made an effort to improve the conclusion in the revised manuscript by making it more concise and suitable.
- The prognosis for liver transplantation in young patients with NASH is poor, according to Alkhouri N [44] et al. This paper differs from the others in that it deals primarily with pediatric cases. The present review deals mainly with adult cases, and it is questionable whether it is appropriate to add the article by Alkhouri N to the discussion. The authors need to consider this point in detail.
Response: We express our gratitude for bringing this to our attention the discrepancy. Following a thorough assessment of the cohort, we identified a significant deviation, and consequently, we have removed the study from the list.
Recently, Jie Ning Yong et al. reported the outcome of liver transplantation for NASH in Clinical Gastroenterology and Hepatology 2023;21:45-54. The authors should show us the difference with this paper.
Response: Thank you for pointing out this paper. In the conclusion of the revised manuscript, we cited this interesting article that we had initially overlooked because we were primarily searching for original articles.
<Minor criticisms>
- The text is littered with unnecessary period after the year, etc. The following period should be omitted.
Response: Thank you for pointing this out. We omitted unnecessary period after the year.
- On page 2, line 49, DM abbreviation needs to be explained.
Response: We explained the DM abbreviation in line 49.
- In the sentence, a period should be a comma. This needs to be corrected.
Response: Thank you for pointing this out. All mentioned period is changed to a comma.
- On page 3 line 121, the hyphen in ‘(UNOS)-’ is not required.
Response: We removed the hyphen in line 121.
- On page 3 line 148, the hyphen in ‘com-plicated’ is not required.
Response: We removed the hyphen in line 148.
- On page 5 line 235, the ‘Dumont-‘ is not required.
Response: We removed ‘Dumont-’ in line 235.
- Please confirm that this is correct for 'NAFLD' on page 5 line 246 and 247.
Response: Thank you for this comment. In the original article, the authors mention NAFLD cirrhosis, but in the revised manuscript we changed it to NASH.
- On page 5 line 254, ')' is required after 'pre-transplant'.
Response: We inserted ‘)’ in the revised manuscript.
- On page 6 line 269, ‘NAFL’ should be ‘NAFLD’.
Response: We inserted ‘MASLD’ in the revised manuscript accordingly to the new nomenclature.
- On page 9 line 433, ‘86’ should be ‘[86]’.
Response: We have corrected the reference tag for number 86.
If you require any additional explanations or details, we are fully prepared to provide them to you. In response to the reviewers' feedback, we have corrected all spelling and grammatical errors. We are eagerly awaiting your response and ready to address any questions or comments you may have.
Sincerely,
Natasa Paklar
Reviewer 2 Report
Comments and Suggestions for Authors
This manuscript is a good update and summary of the LT outlook for people with NASH. It is well organized overall and the outcomes of each study are succinct with the author’s commentary. The following are some suggestions to improve an already good manuscript.
Change the title to severe MALSD since cirrhosis is included in the LT procedures.
Introduction: Times are changing quickly. The AASLD and others came out with new nomenclature in June 2023 and NAFLD/MAFLD are now known as MASLD (metabolic dysfunction-associated steatotic liver disease). MASLD with excess alcohol is now known as MetALD. NASH is now MASH (metabolic dysfunction-associated steatohepatitis) and SLD (steatotic liver disease) is a catch-all term that encompasses all the etiologies of steatosis. https://www.aasld.org/new-nafld-nomenclature. Articles with the title “A multi-society Delphi consensus statement on new fatty liver disease nomenclature” in all the major liver journals should be cited.
This statement should be updated “If the current rate continues, NAFLD will surpass other indications and become the leading cause of LT overall between 2020–2025 [35].” Reference #35 is from 2011 and projects this within our current time period and it is already the latter half of 2023. A more accurate assessment can be made of when LT for MASLD will become the dominant LT procedure.
3. Outcomes: This section seems to be a compilation of studies each with their own paragraph. Do these different studies have titles associated with them? Using the title at the front of the paragraph either in italics or underlined would be helpful to distinguish the different studies cited.
Line 327: “recurrent NASH was detected in one patient (14%), and no recurrent…….” I do not understand the one patient at 14%. Is this a typo or are there missing words?
Line 428: It is seen from studies even periods make a difference. You may want to substitute periods with “time frame” or something similar. Are references 40 and 41 an example of this? If so, can you summarize that outcome for the reader?
Comments on the Quality of English LanguageThere are a few grammatical mistakes that are easily correctable.
Author Response
Thank you for giving us the opportunity to submit a revised draft of our manuscript titled ‘Outcomes of liver transplantation in severe MASLD patients’ to Biomedicines (Special Issue: NASH and Hepatocellular Carcinoma (HCC)). We appreciate the time and effort that you dedicated to providing your valuable feedback on our manuscript. We are grateful for Your insightful comments on our paper. We addressed all the concerns you raised and have been able to incorporate changes to reflect most of the suggestions you provided. We wish to highlight that the new nomenclature was utilized in the revised version of article, in accordance with the agreement reached with the editors. Here is a point-by-point response to your comments and concerns:
- Change the title to severe MALSD since cirrhosis is included in the LT procedures.
Response: Thank you for bringing this to our attention. We have implemented the changes you suggested.
- Introduction: Times are changing quickly. The AASLD and others came out with new nomenclature in June 2023 and NAFLD/MAFLD are now known as MASLD (metabolic dysfunction-associated steatotic liver disease). MASLD with excess alcohol is now known as MetALD. NASH is now MASH (metabolic dysfunction-associated steatohepatitis) and SLD (steatotic liver disease) is a catch-all term that encompasses all the etiologies of steatosis. https://www.aasld.org/new-nafld-nomenclature. Articles with the title “A multi-society Delphi consensus statement on new fatty liver disease nomenclature” in all the major liver journals should be cited.
Response: Thanks for this comment. As the special issue of the journal is named NASH and HCC, that name has been left in the text. Later, in agreement with the editors, we changed the nomenclature.
- This statement should be updated “If the current rate continues, NAFLD will surpass other indications and become the leading cause of LT overall between 2020–2025 [35].” Reference #35 is from 2011 and projects this within our current time period and it is already the latter half of 2023. A more accurate assessment can be made of when LT for MASLD will become the dominant LT procedure.
Response: Thank you for pointing this out. We have come across a recent study that unequivocally demonstrates the current magnitude of the NASH issue.
- Outcomes: This section seems to be a compilation of studies each with their own paragraph. Do these different studies have titles associated with them? Using the title at the front of the paragraph either in italics or underlined would be helpful to distinguish the different studies cited.
Response: After receiving your feedback, we have included a table summarizing the compilation of studies to increase transparency.
- Line 327: “recurrent NASH was detected in one patient (14%), and no recurrent…….” I do not understand the one patient at 14%. Is this a typo or are there missing words?
Response: With only 7 patients, the study is too small to draw statistically significant conclusions. The revised manuscript no longer includes the potentially confusing percentage.
- Line 428: It is seen from studies even periods make a difference. You may want to substitute periods with “time frame” or something similar. Are references 40 and 41 an example of this? If so, can you summarize that outcome for the reader?
Response: Thank you for your message. However, due to varying study periods and overlapping, applying such a division is challenging.
If you require any additional explanations or details, we are fully prepared to provide them to you. In response to the reviewers' feedback, we have corrected all spelling and grammatical errors. We are eagerly awaiting your response and ready to address any questions or comments you may have.
Sincerely,
Natasa Paklar
Reviewer 3 Report
Comments and Suggestions for Authors
Paklar et al. proposed an interesting work concerning NASH and liver transplantation.
Neverteheless, ameliorations are needed:
- References must be updated: NASH treatments are in progress and some molecules were used in recent studies (lanfibranor, liraglutide, ...), NASH is now called MASLD.
- Please, outcome chapter "outcome" would be better in a table than text. The test would be a synthese of the main results.
- Be careful citation 86 is not well placed.
Author Response
Thank you for giving us the opportunity to submit a revised draft of our manuscript titled ‘Outcomes of liver transplantation in severe MASLD patients’ to Biomedicines (Special Issue: NASH and Hepatocellular Carcinoma (HCC)). We appreciate the time and effort that you dedicated to providing your valuable feedback on our manuscript. We are grateful for Your insightful comments on our paper. We addressed all the concerns you raised and have been able to incorporate changes to reflect most of the suggestions you provided. We wish to highlight that the new nomenclature was utilized in the revised version of article, in accordance with the agreement reached with the editors. Here is a point-by-point response to your comments and concerns:
- References must be updated: NASH treatments are in progress and some molecules were used in recent studies (lanfibranor, liraglutide, ...), NASH is now called MASLD.
Response: Thank you for pointing this out. We have, accordingly, revised text to emphasize this point.
- Please, outcome chapter "outcome" would be better in a table than text. The test would be a synthese of the main results.
Response: Agree. We have, accordingly, incorporated a compilation of studies in a table.
- Be careful citation 86 is not well placed.
Response: Thank you for pointing this out. We have corrected the reference tag for number 86.
If you require any additional explanations or details, we are fully prepared to provide them to you. In response to the reviewers' feedback, we have corrected all spelling and grammatical errors. We are eagerly awaiting your response and ready to address any questions or comments you may have.
Sincerely,
Natasa Paklar
Round 2
Reviewer 1 Report
Comments and Suggestions for Authors
This article is a general review of the new outcome of liver transplantation for metabolic dysfunction-associated steatotic liver disease (MASLD), which Dr. Natasa Paklar et al. have completely changed the composition of. This is interesting and very important report for hepatologist and general practitioners.
I have some questions and comments, however, that I believe the authors need to address, as follows:
<Major criticisms>
1. Since Ludwig et al. named non-alcoholic steatohepatitis (NASH) and not metabolic dysfunction-associated steatohepatitis (MASH), we believe that the first part of page 1 should retain the NASH name.
2. The first paragraph on page 1 would be better placed before "Recently," on page 2.
3. The following sentence should be inserted after line 64 on page 2 "Henceforth in this paper, non-alcoholic fatty liver disease (NAFLD) will be reported as MASLD and NASH as MASH."
<Minor criticisms>
1. On page 2 line 61, '()' is not required.
2. On page 10 line 161, ‘live donors’ should be ‘living donor’.
3. Numbers at the beginning of a sentence should be spelled out (on page 10 line191, `5-year` and on page 11 line 248, `95 patients`.
4. On page 10 line 195, the ‘an’ is not required.
5. On page 10 line 198, `They didn’t report overall survival but only disease recurrence.’ is unnecessary because it is redundant.
6. On page 10 line 203, `80%.` this comma should be a period.
7. On page 10 line 210, dose `PSC` need or not, they should confirm this.
8. On page 11 line 228, `from` should be omitted.
9. In this manuscript, ‘et al.’ and ‘at al’ are mixed. We believe they should be unified (with period or without period).
10. On page 11 line 252, in `5,3` this comma should be a period.
11. The []] after the number of the bibliography after page 12 needs to be corrected because it is duplicated.
Author Response
Thank you for taking the time to review our manuscript titled 'Outcomes of liver transplantation in NASH patients' for Biomedicines' Special Issue on NASH and Hepatocellular Carcinoma (HCC). We appreciate the effort you put into providing valuable feedback on our manuscript and we are grateful for your insightful comments. We have addressed all the further concerns you raised and incorporated changes to reflect most of the suggestions you provided. Please find our point-by-point response to your comments and concerns below:
<Major criticisms>
- Since Ludwig et al. named non-alcoholic steatohepatitis (NASH) and not metabolic dysfunction-associated steatohepatitis (MASH), we believe that the first part of page 1 should retain the NASH name.
Response: Thank you for this comment. In this version we have made efforts to enhance our use of naming conventions and we have accepted your suggestion.
- The first paragraph on page 1 would be better placed before "Recently," on page 2.
Response: Thank you for pointing this out. We have revised the opening portion of the text to enhance its clarity and effectiveness.
- The following sentence should be inserted after line 64 on page 2 "Henceforth in this paper, non-alcoholic fatty liver disease (NAFLD) will be reported as MASLD and NASH as MASH."
Response: We appreciate your comment and have taken the liberty to use your sentence.
<Minor criticisms>
- On page 2 line 61, '()' is not required.
Response: We removed '()' from line 61.
- On page 10 line 161, ‘live donors’ should be ‘living donor’.
Response: Thank you for pointing this out. We used phrase ‘living donor’.
- Numbers at the beginning of a sentence should be spelled out (on page 10 line191, `5-year` and on page 11 line 248, `95 patients`.
Response: Thank you for pointing this out. We corrected it.
- On page 10 line 195, the ‘an’ is not required.
Response: We removed ‘an’ from line 195.
- On page 10 line 198, `They didn’t report overall survival but only disease recurrence.’ is unnecessary because it is redundant.
Response: Thank you for pointing this out. We removed it.
- On page 10 line 203, `80%.` this comma should be a period.
Response: We corrected it.
- On page 10 line 210, dose `PSC` need or not, they should confirm this.
Response: After summarizing the section labeled 'Outcome', we noticed that it was not present in the latest version.
- On page 11 line 228, `from` should be omitted.
Response: We removed it.
- In this manuscript, ‘et al.’ and ‘at al’ are mixed. We believe they should be unified (with period or without period).
Response: We corrected it
- On page 11 line 252, in `5,3` this comma should be a period.
Response: We corrected it
- The []] after the number of the bibliography after page 12 needs to be corrected because it is duplicated.
Response: We corrected it
If you need any further explanations or details, we are more than happy to provide them for you. As per the reviewers' feedback, we have rectified all spelling and grammatical errors. We are excitedly looking forward to hearing back from you, and are fully prepared to answer any questions or address any comments you may have.
Sincerely,
Natasa Paklar
Reviewer 3 Report
Comments and Suggestions for Authors
I thank Dr Paklar and al to propose their manuscript with corrections. I want to point nevertheless some important points:
- Be careful with definitions, MASH is not totally equal to NASH. https://www.aasld.org/new-masld-nomenclature. So precise it in the introduction (don't be so long, one line and not in the first line.) the change and the little change. Also, NASH term can be conserved in first histological descriptions and in the related articles. Use MASLD term only when necessary.
Also, introduction structure must be ameliorated.
- Is it necessary to keep a so long texte in chapter "outcomes" if the table is present? A summary can be more useful.
- In epidemiology chapter, you can summarize. Refer to PMID: 35798021.
- Conclusion chapter structure must be clearer.
Author Response
Thank you for taking the time to review our manuscript titled 'Outcomes of liver transplantation in NASH patients' for Biomedicines' Special Issue on NASH and Hepatocellular Carcinoma (HCC). We appreciate the effort you put into providing valuable feedback on our manuscript and we are grateful for your insightful comments. We have addressed all the further concerns you raised and incorporated changes to reflect most of the suggestions you provided. Please find our point-by-point response to your comments and concerns below:
- Be careful with definitions, MASH is not totally equal to NASH. https://www.aasld.org/new-masld-nomenclature. So precise it in the introduction (don't be so long, one line and not in the first line.) the change and the little change. Also, NASH term can be conserved in first histological descriptions and in the related articles. Use MASLD term only when necessary.
Response: Thank you for this comment. In this version we have made efforts to enhance our use of naming conventions.
- Also, introduction structure must be ameliorated.
Response: Thank you for pointing this out. We have revised the opening portion of the text to enhance its clarity and effectiveness.
- Is it necessary to keep a so long texte in chapter "outcomes" if the table is present? A summary can be more useful.
Response: We appreciate your comment and have taken the liberty to summarize it, highlighting only the main points.
- In epidemiology chapter, you can summarize. Refer to PMID:
Response: Thank you for bringing this paper to our attention. We have since referenced and summarized Epidemiology.
- Conclusion chapter structure must be clearer.
Response: Thank you for pointing this out. Therefore, we have made an effort to improve the conclusion in the revised manuscript by making it more concise and suitable.
If you need any further explanations or details, we are more than happy to provide them for you. As per the reviewers' feedback, we have rectified all spelling and grammatical errors. We are excitedly looking forward to hearing back from you, and are fully prepared to answer any questions or address any comments you may have.
Sincerely,
Natasa Paklar
Round 3
Reviewer 3 Report
Comments and Suggestions for Authors
Pashkar and al's corrections are clearer. I won't change the article.
Author Response
Dear Madam/Sir,
We want to express our sincere gratitude for taking the time to revise our article. We truly appreciate the trust you have placed in us, as well as the effort and time you have invested in this process.
We are confident that your invaluable contribution has significantly improved the quality of our work.
Thank you again for your assistance.
Sincerely,
Nataša Paklar